# Automatic Penaeus Monodon Larvae Counting via Equal Keypoint Regression with Smartphones

**DOI:** 10.3390/ani13122036

**Published:** 2023-06-20

**Authors:** Ximing Li, Ruixiang Liu, Zhe Wang, Guotai Zheng, Junlin Lv, Lanfen Fan, Yubin Guo, Yuefang Gao

**Affiliations:** 1College of Mathematics and Informatics, South China Agricultural University, Guangzhou 510642, China; 2South China Sea Fisheries Research Institute (CAFS), Guangzhou 510300, China; 3College of Marine Sciences, South China Agricultural University, Guangzhou 510642, China

**Keywords:** *Penaeus* monodon shrimp larvae counting, highly congested scenes, smartphone, keypoint regression

## Abstract

**Simple Summary:**

To enhance farming efficiency, most farmers prefer to purchase *Penaeus* larvae as opposed to hatching them themselves. However, counting small and highly congested larvae during transactions is a challenging task to accomplish manually. We intend to improve counting precision and decrease human labor costs. In this work, an equal keypoint regression method is proposed to address these challenges. We employed five different types of smartphones to capture thousands of high-resolution images under various challenging environmental conditions. Then, we selected 1420 images to build a high-resolution dataset. In addition, this high-resolution dataset included general point annotations for use. Following training with this dataset, we obtained a model that we tested with a real *Penaeus* monodon larvae dataset. The results showed that the average model accuracy for the 720 images with seven density groups in the test dataset was 93.79%. Ultimately, our trained model demonstrated greater efficiency than the classical density map algorithm.

**Abstract:**

Today, large-scale *Penaeus* monodon farms no longer incubate eggs but instead purchase larvae from large-scale hatcheries for rearing. The accurate counting of tens of thousands of larvae in these transactions is a challenging task due to the small size of the larvae and the highly congested scenes. To address this issue, we present the Penaeus Larvae Counting Strategy (PLCS), a simple and efficient method for counting *Penaeus* monodon larvae that only requires a smartphone to capture images without the need for any additional equipment. Our approach treats two different types of keypoints as equip keypoints based on keypoint regression to determine the number of shrimp larvae in the image. We constructed a high-resolution image dataset named Penaeus_1k using images captured by five smartphones. This dataset contains 1420 images of *Penaeus* monodon larvae and includes general annotations for three keypoints, making it suitable for density map counting, keypoint regression, and other methods. The effectiveness of the proposed method was evaluated on a real *Penaeus* monodon larvae dataset. The average accuracy of 720 images with seven different density groups in the test dataset was 93.79%, outperforming the classical density map algorithm and demonstrating the efficacy of the PLCS.

## 1. Introduction

Aquatic food is a primary source of protein for human consumption, with China currently leading the world in the production, exportation, and processing of aquatic products. China’s share of global aquaculture production is approximately 57% [1]. The production of the *Penaeus* monodon species of shrimp has seen a remarkable increase and reached 104,665 tons in 2021 [2]. The breeding of *Penaeus* monodon requires advanced technology and strict environmental conditions. As a result, shrimp larvae hatching is typically carried out in specialized hatcheries with experienced professionals and dedicated facilities. Farmers purchase shrimp larvae from these hatcheries to continue their aquaculture operations. Estimating the number of larvae, however, remains a challenging and time-consuming task in larvae trading. *Penaeus* monodon larvae have high value but are also relatively fragile and unable to withstand dehydration for extended periods. Prolonged hypoxia can result in the death of larvae [3]. Therefore, shrimp larvae cannot be weighed for estimation, and the quantity must be used as a proxy for the value of the larvae. Nevertheless, the small size of the larvae and their large numbers in transactions make manual counting a labor-intensive process, taking an average of 15–20 min to count 500–700 shrimp larvae [4]. There is an imminent need for an accurate and efficient method for counting shrimp larvae.

Shrimp larva counting methods can be classified into traditional estimation and image processing methods. The traditional estimation includes manual and weighing methods. In the manual method, thousands of larvae are diluted into numerous containers and counted in several batches by multiple individuals. This process is time-consuming, labor-intensive, and harmful to the larvae, as prolonged exposure to air causes stress reactions. In the weighing method [5], the group weight of the larvae is measured and divided by the weight of a single larva to determine the count. This method requires the larvae to be placed in a no-water environment, which is also harmful to their health. Furthermore, the counting accuracy is affected by differences in the size and volume of the larvae.

Image-processing-based shrimp larvae counting methods can be broadly classified into traditional and deep-learning-based approaches. Traditional methods rely on using image-processing techniques, such as thresholding, erosion, dilation, and connected domain analysis, to segment and locate shrimp larvae in images by exploiting the salient features of the hepatopancreas of the larvae. The number of larvae is then estimated based on the connected domains. On the other hand, deep-learning-based approaches utilize target detection and instance segmentation techniques to automatically determine the count of shrimp larvae from static images. However, accurately locating the larvae is challenging due to their overlapping and adhesion, leading to increased error for both traditional and deep-learning-based methods.

Methods that rely on traditional image processing typically require strict control over environmental factors such as image brightness and perspective to ensure consistent results, which often necessitates the use of additional equipment. Khantuwan et al. [6] use Laplacian and median filters to improve the contrast of larvae edges and adaptive thresholding to reduce the effect of nonuniform illumination. The larvae in the image are classified into two groups by taking the 70th percentile of the average area of all connected components of the binary image as the threshold. In the first group, all objects are counted via comparison to the average area of a single larva, which is calculated by the areas of the connected components between the 45th and 50th percentiles. In the second group, all objects are counted via template matching, which scans the whole image. This method sets many fixed parameter values. However, if the larvae sizes are too different or there are impurities in the container, the counting performance will be degraded. Solahudin et al. [7] convert the collected RGB image of shrimp larvae into a grayscale image, use a threshold to separate the shrimp from the background, and then use three dilation operations to connect the shrimp hepatopancreas and shrimp head. This method can only handle dozens of shrimps at a time and requires additional shooting equipment and auxiliary shooting devices, such as industrial cameras, LED fill lights, and transparent platforms. In Ref. [8], the quotient of the original image and the illumination estimation image is taken as the illumination normalized image to solve the problems of uneven illumination and strong illumination of shrimp larvae images. To solve the problem of larval adhesion in the figure, a gradient-marked watershed segmentation algorithm is used for segmentation and morphological processing to improve the segmentation effect. Finally, the connected component method is used for counting. The number of this method is increased to hundreds at one time, but with the increase in the density, the accuracy is significantly reduced. Yeh et al. [9] divide the foreground larvae according to the histogram of V in HSV color space and the larvae threshold range and then uses a K-means clustering algorithm to divide the connected domain into two groups: a single shrimp group and multiple shrimp groups. In multiple shrimp groups, the area of a connected component is divided by the median area of the single shrimp groups and rounded down to obtain the count value. Traditional image processing methods perform well in counting low-density shrimp larvae in specific scenarios, enabling accurate segmentation and counting of shrimp larvae in the images. However, the accuracy of traditional image processing methods significantly declines when environmental factors, such as illumination, change.

Based on the specific principles involved, the methods with deep learning as the core can be divided into target detection, instance segmentation, and crowd counting. There have been some works focusing on detection before counting. Zhang et al. [10] use local images and lightweight YOLOv4 to detect shrimp larvae in aquaculture tanks in real time. This method divides the input image into 16 blocks to improve the resolution of the network input, thus improving the counting accuracy. To meet the requirements of real-time counting, the backbone of the improved YOLOv4 is modified to MobileNetv3. When counting larvae, Lainez et al. [11] first scale the image to 400 × 400 resolution, turn it into a grayscale image, divide it into 16 or 36 images for detection via CNN, and finally summarize the results of each piece. The main contribution lies in the blocking operation of the image, which makes the small objects in the graph more significant and conducive to model learning. The other studies focus on the instance segmentation method to count larvae. Nguyen et al. [12] implement a two-phase Mask R-CNN to segment larvae from the background for counting whiteleg shrimp larvae, which is more accurate than a one-phase Mask R-CNN. The dataset is divided into low, moderate, and high groups according to the degree of overlap of larvae in the image, and the counting performance of the model is analyzed in these three cases. With the increase in the degree of overlap of larvae, the accuracy of model counting is less than 80% in the high group. Hong Khai et al. [13] divide the dataset into low, medium, and high groups according to the larvae quantity in the image and improves upon the Mask R-CNN via a parameter calibration strategy. The maximum number of larvae in the dataset image is 256. In the high group, there are many larvae overlapping each other, resulting in a significant decline in the accuracy of this method. Most of these methods focus on whiteleg shrimp; however, the body shape of the *Penaeus* larvae is elongated, and the widths of the head and tail are almost the same. Moreover, in the highly congested scenarios in the proposed dataset, the shrimp larvae are heavily intertwined, with a maximum of 1691 larvae in a single image.

In addition, some studies have improved the density map counting method in crowd counting to obtain the number of shrimp larvae in static images. Research on congested scenes based on deep learning mainly focuses on crowd-counting analysis. The crowd density in images is very high, and people are heavily blocked from each other. Target detection methods (YOLO [14,15,16,17], Fast R-CNN [18], etc.) are used to count people in images, but the effect is not ideal. Researchers have found a new way to count pedestrians by regressing density maps, and the error of the prediction results of such methods is kept within a reasonable range. Zhang et al. [19] proposed a multichannel convolutional neural network (MCNN), which constructs three parallel convolutional neural networks. The size of the convolution kernels on each path is different. Therefore, the method can accurately obtain the crowd count from a static image with arbitrary crowd density and arbitrary perspective. Li et al. [20] proposed a dilated convolutional neural network (CSRNet) CSRNet that comprises two main components: a front-end convolutional neural network (CNN) for 2D feature extraction and a backend dilated CNN that utilizes dilated kernels to enlarge the receptive fields with higher resolution than the pooling operation, so highly congested scenes can be easily understood. To address the issue of the imperfect generation of ground-truth density maps due to occlusion, varying shooting angles, and changes in person size, Ma et al. [21] proposed Bayesian loss to generate a more accurate density distribution model from point annotations. This method improves the previous research on the generation method of ground truth by using Bayesian loss to more reliably supervise the count expectation at each annotated point to generate a more reasonable density map. CCTrans [22] is a high-performance crowd-counting network based on a vision transformer. It has demonstrated exceptional performance with multiple crowd-counting datasets. However, due to the use of a vision transformer as the backbone network, there are strict requirements for the input. The input needs to be processed into specific sizes, which may result in the loss of some detailed features. Density-map-based counting methods mainly focus on crowd-counting scenarios and cannot provide accurate localization of individual objects.

Fan et al. [23] improved the density map method, applied it to the field of shrimp larvae counting, and constructed a shrimp larvae dataset. Compared with excellent density map counting methods such as MCNN, CSRNet, and CAN [24], the accuracy is greatly improved. However, due to the limitations of the density map method, the prediction results have difficulty providing reliable shrimp larvae prediction position coordinates, which is not conducive to the subsequent performance analysis of the algorithm. In addition, additional equipment is needed to ensure consistent image quality, and the operation process is more complex. Wang et al. [25] used an improved network that added the improved spatial attention mechanism (SAM) module in the feature map fusion stage of the UNet [26] network as the backbone network of the density map counting method. This method tested in the DIou_Shrimp dataset is significantly better than the classical density map method, and the experimental results show that the method can alleviate the problems of occlusion and adhesion in larvae. Although the counting accuracy of this method is greatly improved, the calculation efficiency is reduced due to the SAM module.

To address the above issues, Figure 1 illustrates the automatic Penaeus monodon larvae counting method via equal keypoint regression to obtain the number of larvae in images captured by only smartphones. The proposed method adopts two different types of keypoints as one equal type of keypoint. Then, equal keypoint positions can be obtained via the PLCS from the heatmap output of the backbone. Therefore, the counting result of larvae in the images can be easily obtained.

The main contributions of this study can be summarized into three parts. First, we collected a large number of images of *Penaeus* monodon larvae without any other instrument assistance, constructed a large dataset Penaeus_1k, and carried out multi-keypoint annotations to verify the shrimp larvae counting performance based on deep learning. Second, we introduce a *Penaeus* monodon larvae counting method, which can not only accurately predict the number of larvae in the image but also accurately locate the keypoints of larvae in the image. Finally, extensive experiments are conducted on the Penaeus_1k dataset to demonstrate the effectiveness of the proposed method.

The rest of this paper is organized as follows. Section 2 presents the collection process of the Penaeus_1k dataset and the details of the proposed method. The experiments and applications are presented in Section 3, and finally, the discussion and the conclusion are presented in Section 4 and Section 5, respectively.

## 2. Materials and Methods

### 2.1. Materials

#### 2.1.1. The Collection of Raw Larvae Data

To standardize the collection process, a suitable volume of water that can fully cover the larvae is added to the container. A small number of larvae is then added to the container as the first group, followed by seven additional groups, each containing successively more larvae. To ensure consistency, five individuals take turns collecting 50 images after each addition of larvae. The image collector holds the mobile phone at a distance of 300 mm to 500 mm above the container and captures images that include all of the larvae in the container while avoiding any background outside of the container as much as possible. Table 1 describes the specification of capture devices used by five individuals. To diversify the dataset, images are collected from different perspectives, resulting in variations in the background, perspective, and light intensity. The water is gently stirred occasionally during the collection process to prevent the larvae from clustering and to obtain a different distribution of larvae in the same group.

#### 2.1.2. The Penaeus_1k Dataset

This acquisition task collected 2000 JPEG RGB images of *Penaeus* monodon larvae. According to the image acquisition batch, the images are divided into eight density groups. There are 250 images in each group. The number of larvae is almost identical in the images in the same group. The larvae do not connect to each other in groups (1~5) so that the model can better learn the characteristics of keypoints of larvae. There are a large number of larvae in groups (6~7), which require a large amount of work and time to annotate. Therefore, the images selected for the training set are mainly groups (1~5) with 108 images for each group and 10 images for each group (6~7) to fine tune the model to improve the prediction performance of the model for congested scenes. Since the training set has been annotated in the corresponding group, the number of larvae in each group is known, so the validation set and test set do not need to be annotated again. The proportion of training set and validation set images is 8:2. There are seven groups with 560 images in the training set, and there are seven groups with 140 images in the validation set, with 20 images in each group. A group of ultraheavy larvae images was included in the test set to evaluate the counting performance of the model under highly congested scenes. The eight groups of testing groups contain 90 images each, totaling 720 images. It is worth noting that the images are randomly selected from the original images, and the images in each dataset do not coincide. Finally, 1420 diversified images are selected to construct the dataset. These images also contain a variety of different image features to train and test the stability of the model under the influence of different factors.

The dataset is annotated by Labelme v5.0.1 software with point annotation. The annotation method identifies the heads, abdomens, and tails of the larvae to obtain the coordinates of each keypoint. In scenarios with high density, certain shrimp larvae keypoints may be obstructed by other larvae. In such cases, it is necessary to manually estimate the position of the obscured keypoints for annotation instead of disregarding them. All images in the training set need to be annotated, while those in the validation set do not, and the test set only needs to be annotated with an ultraheavy larvae image to obtain the number of larvae in the image. After the annotation tasks are completed, the images are assigned to other team members for cross-validation. If there are instances of multiple annotations, missed annotations, significant deviations in the keypoint positions, or the annotated results for the same group differ by more than five individuals, the annotations will need to be revised and resubmitted. For images with extremely high density, multiple team members conduct sequential checks to ensure annotation quality and accuracy in counting the total number of shrimp larvae in the images. Table 2 shows the number of larvae for a single image in each group obtained by annotating the image. Finally, the annotation file generated via LabelMe is converted to the widely used COCO dataset format in deep learning, enabling the utilization of various models for training and testing on this dataset.

Figure 2 shows some images in the dataset. There is a light reflection in Figure 2a,c,e and uneven light intensity in Figure 2a–g, which result in differences in image brightness. Figure 2g belongs to an additional image group added to the test set that is not in the training set. It tests the performance of the model under extremely dense larvae. There are approximately 1691 larvae in the image, and there is a serious overlap between shrimp larvae, which is quite challenging and meaningful for testing. In addition, the images are obtained from an arbitrary perspective and contain a varying number of larvae, which is similar to the complex and arbitrary usage scenarios.

### 2.2. Methods

An overview of our method is shown in Figure 3. In the training stage, a backbone feature extraction network is used to extract the predicted heatmap. Here, following IIM [26], we use HRNET [27] as the base network. The MSE (mean-squared) loss is used to measure the difference between the prediction results and ground truth. In the testing stage, the predicted heatmap is obtained based on the trained counting model, and the number of larvae and locations of the keypoints are obtained by the proposed Penaeus Larvae Counting Strategy (PLCS) from the output of the backbone.

#### 2.2.1. Ground Truth Generation

The collected larvae images are annotated by point annotations. During training, the point annotations need to be converted into the corresponding heatmaps as the ground truth. According to ablation Section 3.4, we choose the heads and tails of the larvae to generate the ground truth. The annotations are saved in the annotation files in the form of coordinates, which greatly reduces the size of the files saved. However, the coordinate form information is not suitable for generating the ground truth. Therefore, it is necessary to first convert the coordinate annotations into an annotation matrix with the same size as the corresponding image. In the annotation matrix, the value of an annotated pixel is 1, and the value of a nonannotated pixel is 0. This process allows us to obtain the annotation matrix H(x) for the image, which represents the binary map of annotated keypoints. Next, the Gaussian kernel is required to convolve the annotation matrix to obtain the ground truth, as shown in Equation (4). For the Gaussian kernel generation mode, refer to Equations (1)–(3).
(1)s=6 ∗ σ+3
(2)$$x_{0}=3\;*\;\sigma +1,\;y_{0}=3\;*\;\sigma +1$$
(3)$$g=e^{-\frac{{\left(x-x_{0}\right)}^{2}+{\left(y-y_{0}\right)}^{2}}{2\;*\;\sigma^{2}}}$$
(4)$$F\left(x\right)=H\left(x\right)\;*\;G_{\sigma }\left(x\right)$$

The larvae in the images collected by smartphones are similar in size, and σ is set to 3 to obtain a Gaussian kernel with a fixed width and height of s pixels. The value range of y is 0~s, and x_0_ and y_0_ are the central position coordinates of the Gaussian kernel calculated using Equation (2). According to the two-dimensional Gaussian function Equation (3), we obtain a Gaussian kernel that follows the two-dimensional Gaussian distribution. Then, the Gaussian kernel is used to convolve the annotation matrix, as shown in Equation (4), and when there are overlapping regions in the results, we choose the maximum value as the ground truth.

The original, annotated, and ground truth images are shown in Figure 4. The Gaussian kernel parameter *σ* is set to 3. The keypoints of the head and tail are used to generate the ground truth.

#### 2.2.2. Feature Extracting Module

Many existing deep learning network methods [28,29,30,31] first downsample the input image gradually to obtain high-dimensional but low-resolution feature representations and then use transpose convolution or other methods to conduct an upsampling operation on the feature representation to restore the resolution. High-dimensional features represent higher-order image semantic information, which is more conducive to image classification. However, a simple upsampling operation makes the feature representation lose more details, which easily leads to inaccurate positioning. Low-dimensional features have higher resolution but weak semantic information and more noise. If high-dimensional and low-dimensional features can be efficiently fused, strong semantic and high-resolution feature results can be generated.

We use HRNet-w48 as the feature extraction module, as shown in Figure 5, and the network configuration follows HRNet-w48. HRNet-w48 consists of four stages, 1, 2, 3, and 4, which contain 1, 1, 4, and 3 exchange blocks, respectively. After each stage is executed, the result of fusing the high-resolution branch with the downsampled result of the low-resolution branch is used as the input for the lowest-resolution branch of the next stage. There are a total of four branches. HRNet completes the fusion of high-dimensional features and low-dimensional features via repeated multi-resolution branch information interaction to produce richer and more accurate results. The higher resolution branch reduces the resolution through a downsampling operation, while the lower resolution branch improves the resolution using the nearest neighbor interpolation upsampling method. In this study, the output of branch 1 with the highest resolution is used as the heatmap of the model output for further localization and counting analysis.

#### 2.2.3. Penaeus Larvae Counting Strategy

After extracting features from the backbone network, a heatmap with the same size as the input image is obtained. The heatmap contains rich information. The value of each pixel represents the confidence that the pixel is the corresponding keypoint. The higher the confidence is, the higher the probability that the pixel is the corresponding point. To obtain the position of larval keypoints, it is necessary to obtain the maximum pixel value of the local area. The local maximum acquisition strategy is shown in Algorithm 1.

First, 3 × 3 maximum pooling is used for nonmaximum suppression to obtain the pixels (candidate points) with the highest confidence in the local area. Then, we use the threshold to filter out false positives and obtain true positives. The experiment shows that the pixel value of true positives is far greater than that of false positives, which means that if the pixel value of a point is large enough, it is more likely that the point is a keypoint of larvae. Then, the threshold Th is used to filter out false positives to obtain the keypoints of larvae. E_num_ is the number of equal keypoints. Following Section 2.2.4, E_num_ is set to 2.
**Algorithm 1** Penaeus Larvae Counting Strategy**Input:** predicted heatmap ***H*** generated by backbone.**Output:** coordinates ***C*** and quantity ***Q*** of the larvae in the input.     /* {*Boolean* ? *A*:*B*} means returning *A* if it was **true**, otherwise *B* */     /* get all candidate points */*dilate_matrix* := *maxpooling(H, size = (3,3))***For** *i, j* **in** *dilate_matrix*:*candidate_points[i][j]* := **{***dilate_matrix[i][j]* == *H[i][j]*
**?**
*H[i][j]***:0}**/* utilize a threshold ***T_h_*** to filter the false positives */**For** *i, j* **in** *candidate_points*:*candidate_points[i][j]* := **{***candidate_points[i][j]* < ***T_h_* ? 0:1}**/* obtain the coordinates and quantity */***C*** = nonzeros*(candidate_points)****Q*** = ⌊ sum*(candidate_points)*/*E_num_* ⌋**Return *C*, *Q***

#### 2.2.4. Experimental Setup

To expand training samples and inhibit model overfitting, we use data augmentation techniques in training, including random clipping, random rotation, random horizontal flipping, etc. Due to the small size of the shrimp, to ensure the high resolution of the image and lower computational complexity, the image pixel values input into the network were scaled to 1024 × 1024 during training.

The σ of heatmap generation is set as 3, the number of training epochs is 50, the batch size was 4, and the size of the heatmap is 256. We use Adam to optimize the model with a learning rate of 0.0015. A Graphical Processing Unit (GPU) and Compute Unified Device Architecture (NVIDIA CUDA) are used to accelerate the model training. The details of the training parameter settings are described in Table 3.

The proposed method is a counting method based on locating keypoints, so the more accurate the locating of larvae keypoints, the more accurate the larvae counting results. Therefore, the loss function is MSE loss (the mean squared error) to compare the difference between the ground truth generated in Section 2.2.1 and the heatmap output of the backbone. MSE loss is shown in Equation (5).
(5)$${MSE}_{loss}=\frac{1}{2N}\sum_{i=1}^{N}{\left\|F\left(X_{i};\sigma \right)-F_{i}\right\|}^{2}$$

#### 2.2.5. Method Performance Evaluation

There are several metrics to evaluate the performance of the proposed method. MAE and MSE are used to measure the error of larvae counting. If the values of MAE and MSE are smaller and the accuracy is higher, the counting result is better. The equations are as follows:(6)$$Acc=\frac{1}{N}\sum_{i=1}^{N}(1-\frac{\left|{Pred}_{i}-{Gt}_{i}\right|}{{Gt}_{i}})\times 100\%$$
(7)$$MAE=\frac{1}{N}\sum_{i=1}^{N}\left|{Pred}_{i}-{Gt}_{i}\right|$$
(8)$$MSE=\sqrt{\frac{1}{N}\sum_{i=1}^{N}{\left({Pred}_{i}-{Gt}_{i}\right)}^{2}}$$
where Pred is the predicted number of larvae, Gt is the actual number of larvae in the image, and N is the total number of images.

### 2.3. A Smartphone App for Shrimp Larvae Counting

The proposed algorithm for automatically counting shrimp larvae is implemented through a WeChat applet, which offers a concise and convenient user interface. Figure 6 illustrates the usage of the app. On the applet’s homepage, users can click the green count button situated in the middle. They have the option to either capture a photo or select a shrimp larvae image from their smartphone album. Subsequently, the user can upload the image to the cloud server for processing. On the cloud server, the mask R-CNN algorithm is initially employed to extract the image container, thereby eliminating background interference. To enhance the resolution of the images input into the backbone network, the picture is divided into four segments. These segments are then individually sent into the backbone network, and PLCS is applied to the output feature map to obtain the counting results and accurately determine the positioning of the shrimp keypoints. Finally, the shrimp larvae counting results, along with the processing time, original image, and labeled image, are delivered to the result page for display. Each counting result is saved in the counting history for future review.

## 3. Results

### 3.1. Model Training Results

The proposed method was trained on the Penaeus_1k dataset. The training loss is shown in Figure 7. At the beginning of training, the loss was at a high level. However, as the number of training iterations increased, the loss rapidly decreased and stabilized after 20 epochs, reaching a very low value. The rapid decrease in loss indicates that the network was able to quickly learn the image features, further confirming the validity of the model. Additionally, training the model for 50 epochs on a GeForce RTX 3090 was completed in only 72 min.

### 3.2. Image Counting Results

The performance of the model in counting shrimp larvae needs to be thoroughly evaluated using a range of indicators. In the Penaeus_1k test set, we employed Acc, MAE, and MSE as evaluation metrics for the proposed method. Table 4 presents the average counting performance of the model in a test set composed of 720 images. The average accuracy of larvae counting in seven different groups (53, 183, 312, 427, 510, 675, and 883) is 93.79%, and the MAE is 33.69, which indicates an average counting error of 33.69 in 720 images across the seven groups. The MSE, which is more sensitive to abnormal values, is 34.74, reflecting the stability of the model; a lower value indicates a more stable model with fewer instances of large counting errors.

Figure 8 displays the average counting performance of the proposed method across different larvae density groups in the test set. As depicted in Figure 8a, as the density of larvae increases, the accuracy of the model’s counting decreases, while both the MAE and MSE increase. In Figure 8b, it can be observed that Group 53 exhibits the lowest MAE and MSE; however, its accuracy is not as high as that of Group 183. This can be attributed to the calculation method of accuracy, where even a minor counting error in low-density groups (such as Group 53) can significantly impact the accuracy of the method. For instance, if the ground truth count is 53 and the model predicts 50, then the accuracy would be approximately 94.4%.

The performance of the method in each group of the test set is visually illustrated in Figure 9. For Groups 53 to 883, the number of shrimps increases continuously, and the occlusion among them becomes more prominent. Nevertheless, our method maintains an accuracy of over 90%, as shown in Figure 9a–d. However, for Group 1691, in the rightmost image of Figure 9l, the rectangle indicated by the arrow represents a case of keypoint detection failure. there is significant crowding, occlusion, and overlapping of shrimp in a large portion of the image. In such cases, it is nearly impossible to accurately distinguish occluded shrimp manually, and occlusion also hinders the accurate localization of keypoints, leading to further degradation in terms of the counting performance. The average accuracy for this group is only 75.69%. In addition to the variations in shrimp density and occlusion among shrimp, the Penaeus_1k dataset also includes a substantial number of images captured from multiple viewpoints, as depicted in Figure 9e–h. Multiple perspective images introduce certain distortions in shrimp morphology, but our proposed method can handle such situations stably, with only a slight decrease in accuracy compared to that from images captured parallel to the container. Furthermore, in real-world application environments, lighting conditions are highly complex, as demonstrated in Figure 9i–k, showcasing images with intricate lighting conditions. The histogram is displayed in the upper right corner of the image, while the brightness mean and standard deviation are shown at the bottom. The histogram illustrates the distribution of brightness levels, the mean represents the average brightness of the image, and the standard deviation partially measures the image’s contrast. It can be observed that the majority of pixels fall within the high brightness range, but there are still many pixels with low brightness. In the image, this is manifested as predominantly bright regions with some darker areas, as well as the presence of glare and halos. Our method can effectively handle the uneven brightness in images and detect shrimp under halos, maintaining a high level of accuracy.

Therefore, to achieve optimal performance with our method, it is crucial to enhance the quality of the images, including maintaining uniform brightness, improving contrast, enhancing image clarity, ensuring the complete presence of the container in the image, and avoiding excessive crowding of shrimp. We recommend keeping the number of shrimp larvae counted in a single image between 500 and 800.

### 3.3. Comparisons with the Crowd Counting Methods

Currently, deep-learning approaches utilizing point annotation datasets are prevalent in the field of crowd counting, and our method is compared against three density map counting methods [20,21,22]. The three methods are all tested on Penaeus_1k. The results of the comparison are shown in Table 5 and Figure 10.

In comparison to those of other density map counting methods, such as CSRNet, CCTrans, and BLNet, our method exhibits significant improvements in terms of MAE and MSE across all groups. Specifically, the proposed method achieves a reduction of 13.73 in MAE and 13.03 in MSE, while also achieving a notable increase of 7.66% in accuracy. The proposed method exhibits significant improvements compared to CCTrans for Group 53, i.e., the low-density group. Specifically, the MAE and MSE decreased by 16.05 and 16.40, respectively. Moreover, there is a remarkable increase in accuracy of 30.28%. The reason for the poor accuracy of density map methods can be observed in Figure 11. In low-density scenarios, density map methods tend to overestimate the number of shrimp larvae. Additionally, due to the accuracy calculation method, even minor errors in shrimp larval counting during low-density situations can significantly decrease the accuracy.

### 3.4. Ablative Analysis

The body shape of *Penaeus* larvae is elongated and slender, allowing for clear differentiation of three types of keypoints from the image: the head, abdomen, and tail. Thus, the annotation data for each shrimp larva consists of three keypoints, representing the head, abdomen, and tail, as shown in Figure 4b. The accuracy of the model predictions varies with different combinations of keypoints. In this section, we investigate the impact of different keypoint combinations on the accuracy of the model. Each shrimp larva has three annotated keypoints, from which all or some can be selected for combination.

The proposed method is trained with the same parameters as detailed in Section 2.2.4 and evaluated on the test set. Figure 12 illustrates the performance of the seven different keypoint combinations on the test set, while Table 6 presents the average accuracy, MAE, and MSE of these models. The red curve indicates the accuracy of Group 2, and the other colors show the accuracy of the other groups (see the legend for details). Notably, the head–tail keypoint combination outperforms other methods in accuracy at each density level and is also more stable. This result can be attributed to the special slender body shape of the *Penaeus* larvae. Hence, we employed the head–tail keypoint combination to generate the ground truth for the proposed method.

### 3.5. On BBBC041v1 Dataset

Beyond the *Penaeus* monodon larvae counting, we conducted an experiment on the BBBC041v1 dataset obtained from the Broad Bioimage Benchmark Collection [32] to evaluate the performance of the proposed method in cell counting tasks, showcasing the robustness and generalization. The dataset consists of 1364 images categorized into six classes, comprising approximately 80,000 cells. All cells in the dataset are annotated using bounding boxes. We have selected red blood cells as the counting objects due to their highest proportion. To obtain the ground truth, we extract the center points of the bounding box annotations of red blood cells using a script. These center points serve as keypoints for the red blood cells.

The evaluation metrics used in previous methods [33,34,35,36,37] on this dataset are different from the metrics proposed in Section 2.2.5, as they primarily focus on metrics related to object detection or instance segmentation. In our study, we utilized a point-based dataset generated from the BBBC041v1 to retrain BLNet, CCTrans, and the proposed method. The only parameter that was adjusted was the Enum parameter of the PLCS algorithm, which was set to 1. All other parameters remained consistent with the ones described in Section 2.2.4. After testing on the test set, we obtained the results presented in Table 7. Our proposed method achieves an accuracy of 82.33% on the test set. Compared to CCTrans, our method improves the accuracy by 0.12%, reduces the MAE by 0.84, and decreases the MSE by 3.04. These results demonstrate that our method not only enhances accuracy but also increases stability, maintaining a high level of performance.

## 4. Discussion

During shrimp transactions, a significant amount of time is spent on shrimp counting, which severely affects the efficiency of shrimp trading. In this paper, we propose an efficient shrimp counting method based on PLCS and HRNet-w48. The proposed method accurately locates the keypoints of shrimp larvae and simultaneously obtains the number of shrimp larvae in the images. Our method, in contrast to traditional image-processing methods, does not require the use of additional equipment to ensure consistency in the image acquisition environment. However, it is important to note that the counting accuracy of the proposed method can be significantly affected by poor image quality. Other methods, such as density map methods, are unable to locate the keypoints of shrimp larvae, limiting further analysis and application of the model’s correctness. In contrast, our proposed method can accurately locate the keypoints of shrimp larvae. However, due to the small size of shrimp larvae, the correspondence between the keypoint representing the head and the keypoint representing the tail cannot be determined. In future research, we plan to explore grouping techniques for the keypoints of shrimp larvae. One potential application is to measure the distance between the head and tail keypoints to study the measurement of shrimp larvae body length. This will contribute to addressing the issue of accurately measuring the body length of shrimp larvae.

To facilitate automatic shrimp counting, we developed a system based on the proposed deep-learning method and smartphones. The system utilizes smartphones to capture images of shrimp larvae, which are then uploaded to a server for processing using the deep-learning algorithm to count the number of shrimp larvae in the images. The server returns the counting results and visualized images. Due to the small size and large quantity of shrimp larvae, high image resolution is required, resulting in large image file sizes, longer network transmission times, and increased model inference time. Through multiple system validation tests, the average counting time ranges from 5 to 10 s (with fluctuations due to network conditions), while the model inference time is approximately 1 s. In the future, we aim to reduce image resolution while maintaining a high level of accuracy, thereby further improving the counting efficiency of the system.

## 5. Conclusions

This study proposes a user-friendly, efficient, and accurate counting method for black tiger shrimp larvae. This method can accurately estimate the number of larvae in the container, improve the efficiency of black tiger shrimp larvae transactions between the hatchery and farmers who rear young shrimp, help farmers more accurately know the number of shrimp larvae purchased, and make planning for the subsequent breeding process more reasonable. After an extensive data collection and annotation period, we developed a dataset called Penaeus_1k of *Penaeus* monodon shrimp larvae. For feature extraction, we employed the backbone HRNet-w48, which can preserve high resolution while incorporating semantic information to enhance counting and localization performance. Subsequently, we introduced a simple counting head, PLCS, to process the feature maps generated by the back-bone network and obtain accurate counting results.

The proposed method was evaluated on the Penaeus_1k test set, which achieved favorable counting results. The average accuracy rate reached 93.79%, with an MAE of 33.69 and an MSE of 34.74. In comparison to density map methods, our proposed approach demonstrates superior overall performance. It not only accurately determines the larvae keypoint positions but also exhibits high accuracy and stability.

## Figures and Tables

**Figure 1 animals-13-02036-f001:**
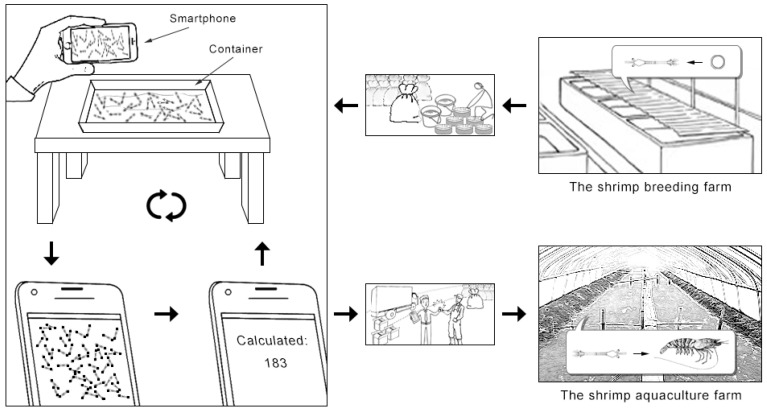
Automatic counting system for *Penaeus* larvae. Our method can help shrimp farmers quickly estimate the number of shrimp larvae, reduce shrimp larval transportation time, and improve the shrimp larval survival rate.

**Figure 2 animals-13-02036-f002:**
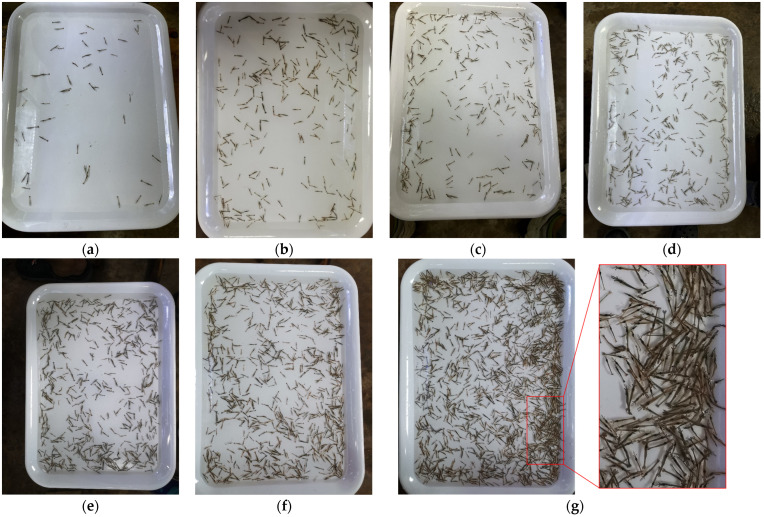
(**a**–**g**) are examples of raw larvae data, which are challenging due to dramatic viewpoints, occlusion, background complexity, and uneven brightness.

**Figure 3 animals-13-02036-f003:**
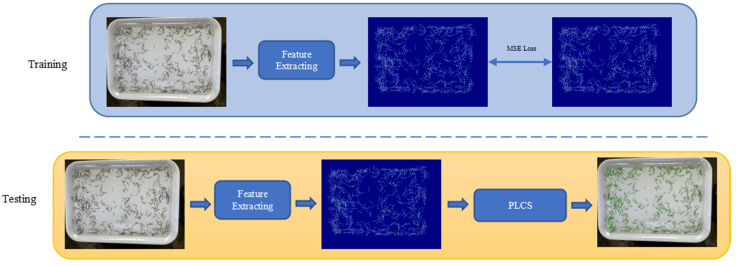
The pipeline of our method. During the training phase, the MSE loss is adopted. During the testing phase, each equal keypoint can be obtained via the PLCS, and the final count is equal to the number of local maxima divided by the number of keypoint types.

**Figure 4 animals-13-02036-f004:**
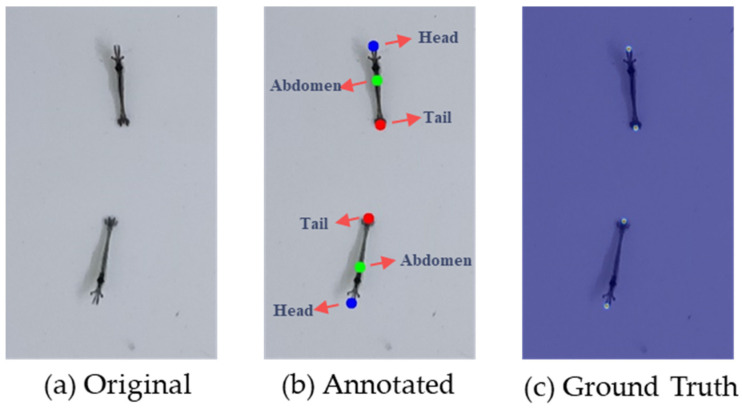
The original, annotated, and ground truth images. We annotated three kinds of keypoints on each larva: head, abdomen, and tail. The head and tail are regarded as the same kind of keypoint to generate the ground truth.

**Figure 5 animals-13-02036-f005:**
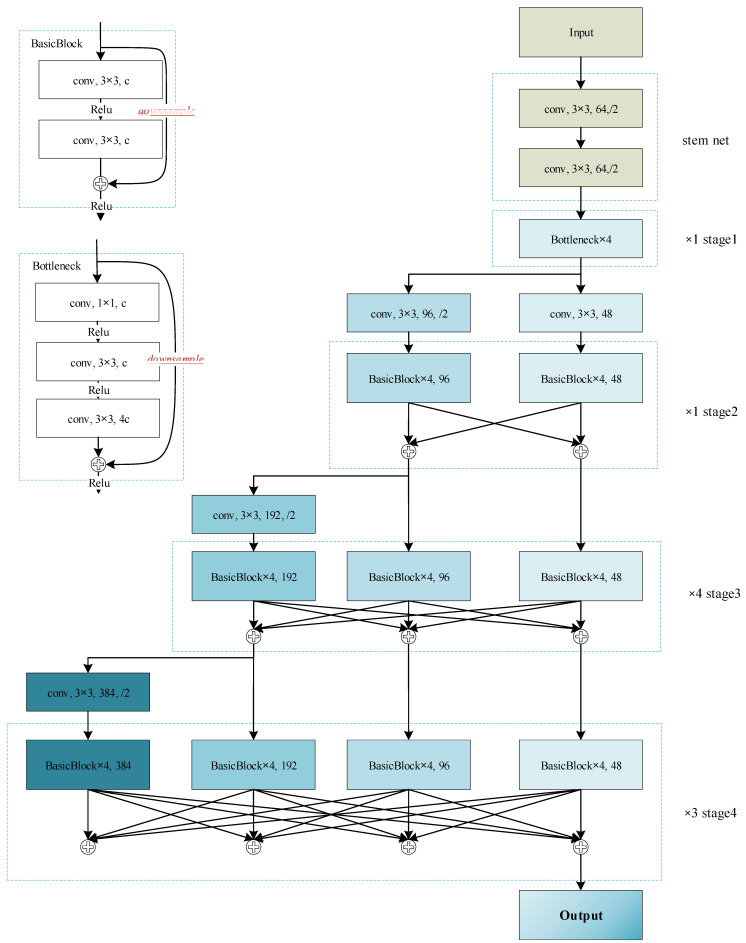
Illustration of the backbone of the proposed method. The output of branch 1 with high resolution was used as the input of PLCS.

**Figure 6 animals-13-02036-f006:**
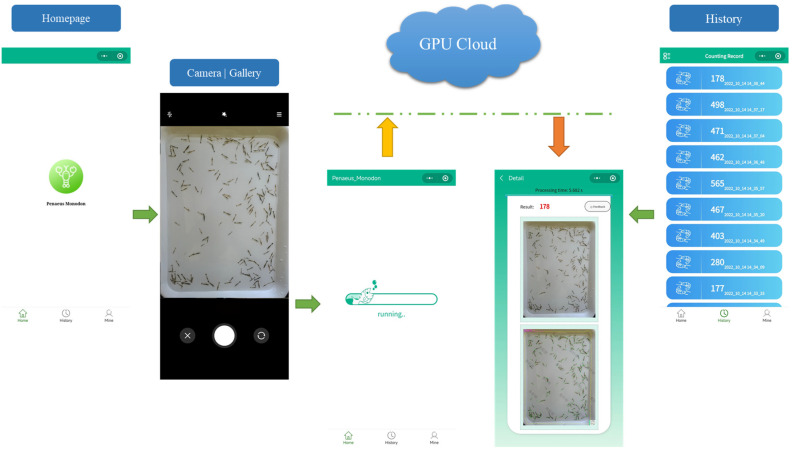
Schematic diagram of the shrimp larvae counting platform. The images are uploaded to the GPU Cloud via the user’s smartphone network. Due to the small size and large quantity of shrimp larvae, it is essential to ensure that the images are as clear as possible to improve counting accuracy.

**Figure 7 animals-13-02036-f007:**
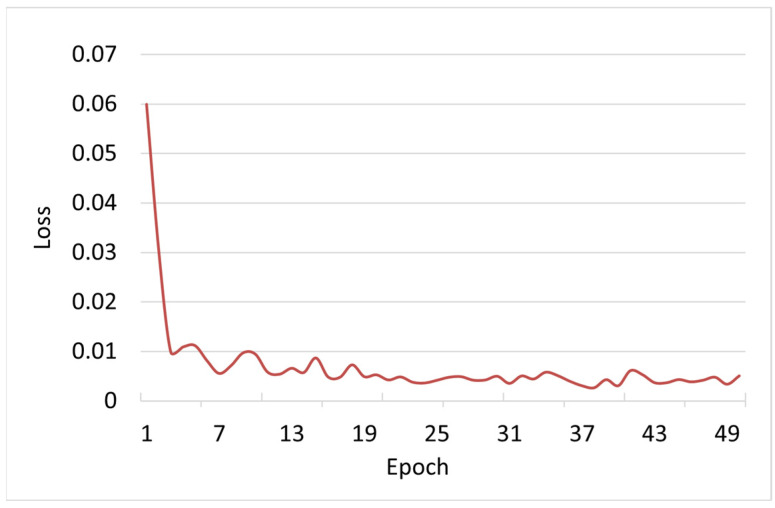
Training loss function value curve. The loss decreases rapidly as the number of training iterations increases.

**Figure 8 animals-13-02036-f008:**
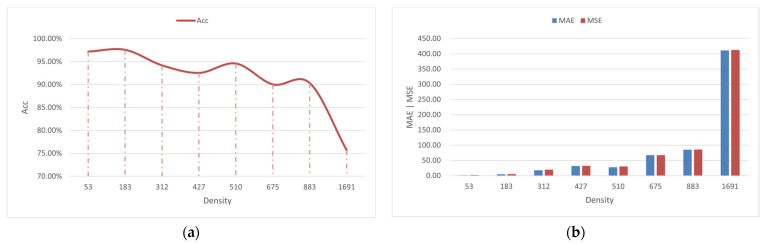
Average performance of the proposed method in eight groups. (**a**) Accuracies with different densities; (**b**) MAEs and MSEs with different densities.

**Figure 9 animals-13-02036-f009:**
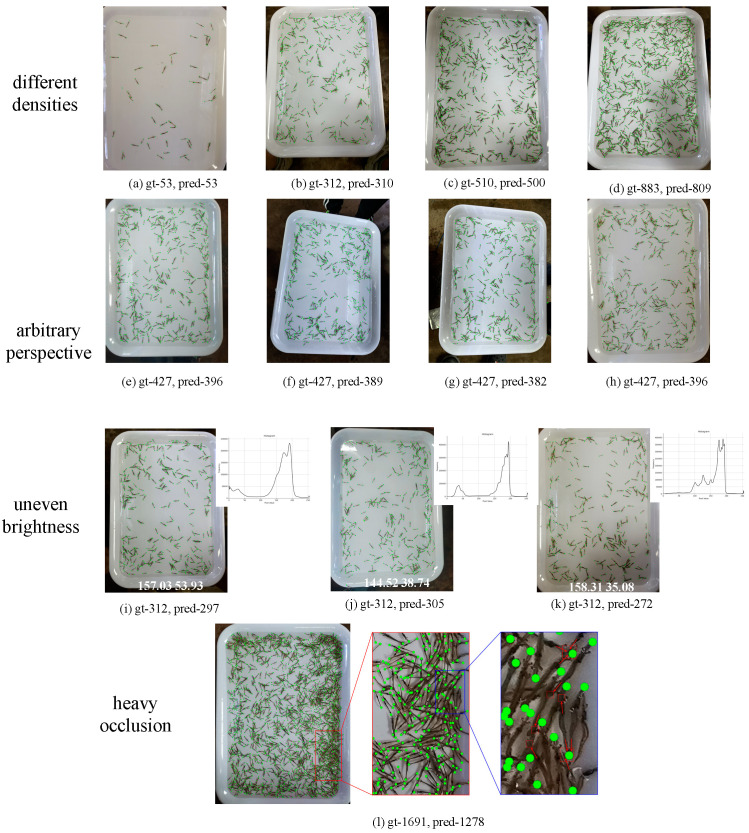
The prediction and localization results of the proposed method under different densities, arbitrary perspectives, uneven brightness, and heavy occlusion. Despite various challenging environmental conditions, the method still accurately locates and provides reliable counting results.

**Figure 10 animals-13-02036-f010:**
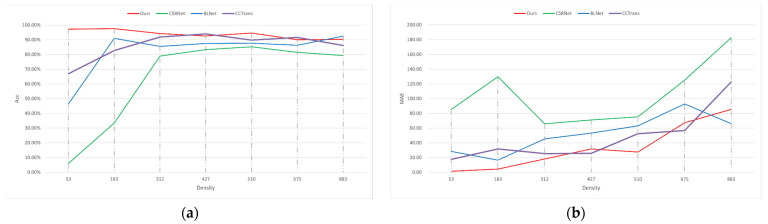
The accuracy and MAE vs. density curves produced by four different counting models on the testing set. The results of other methods are obtained by training and testing using the official code on our dataset. (**a**) Accuracy comparison; (**b**) MAE comparisons.

**Figure 11 animals-13-02036-f011:**
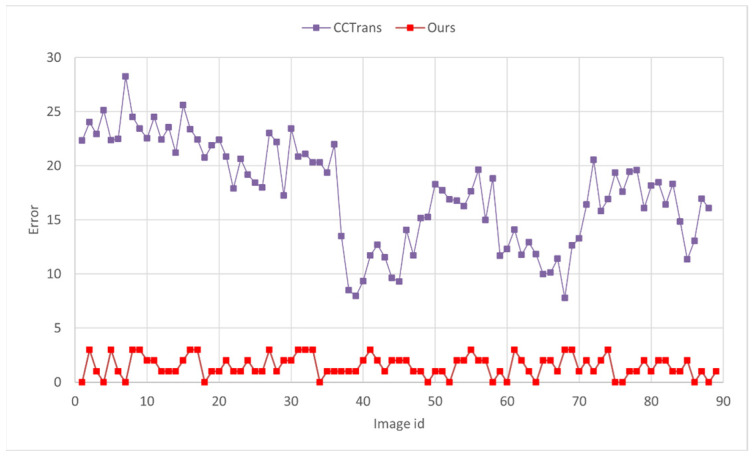
The difference between the predicted values and the ground truth for the low-density group, i.e., Group 53, is shown for both the density map method and our proposed method.

**Figure 12 animals-13-02036-f012:**
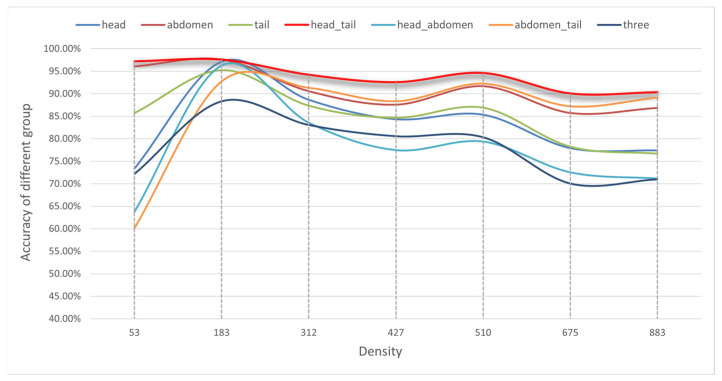
The accuracy of models of different keypoint combinations.

**Table 1 animals-13-02036-t001:** The specification of capture devices. We utilized the smartphone’s default automatic shooting mode to capture images, thereby streamlining the operation process. It is worth noting that images produced by different devices can exhibit variations in factors such as image resolution and image quality.

Device Model	Main Cameras	Image Resolution	Shooting Mode
iPhone 11	12 MP (wide), 12 MP (ultrawide)	4032 × 3024	Auto
iPhone 13	12 MP (wide), 12 MP (ultrawide)	4032 × 3024	Auto
Redmi K40	48 MP (wide), 8 MP (ultrawide), 5 MP (macro)	3456 × 4608	Auto
Huawei P20	12 MP(wide), 20 MP(wide)	2736 × 3648	Auto
Huawei P50 Pro	50 MP (wide), 64 MP (periscope telephoto), 13 MP (ultrawide), 40 MP (B/W)	3072 × 4096	Auto

**Table 2 animals-13-02036-t002:** The number of larvae for a single image in each group.

Group	The Number of Larvae
1	53
2	183
3	312
4	427
5	510
6	675
7	883
8	1691

**Table 3 animals-13-02036-t003:** Training parameter settings.

Parameter	Value
Epoch	50
σ	3
Batch size	4
Input size	1024
Heatmap size	256
Optimizer	Adam
Learning rate	0.0015

**Table 4 animals-13-02036-t004:** Evaluation metric results of the proposed method.

Metric	Result
Acc	93.79%
MAE	33.69
MSE	34.74

**Table 5 animals-13-02036-t005:** Average of metric comparisons with different counting methods on the Penaeus_1k dataset.

Method	Accuracy (%)	MAE	MSE
CSRNet	63.94	105.00	126.61
BLNet	82.42	52.18	62.75
CCTrans	86.13	47.42	58.33
Ours	93.79	33.69	45.30

**Table 6 animals-13-02036-t006:** Seven combinations of keypoints, including one type of keypoint, two types of keypoints, and three types of keypoints. It can be observed that the combination of head and tail keypoints yields the highest accuracy, as well as the lowest MAE and MSE values.

Group	Keypoints	Accuracy (%)	MAE	MSE
1	head, abdomen, tail	77.95	104.28	108.59
2	head, tail	**93.79**	**33.69**	**34.74**
3	head, abdomen	77.73	102.69	105.53
4	abdomen, tail	85.88	47.56	53.13
5	head	83.44	77.91	80.50
6	abdomen	90.87	49.11	50.76
7	tail	84.99	77.18	79.78

**Table 7 animals-13-02036-t007:** Different counting methods were evaluated on the BBBC041v1 dataset, and the proposed method achieved the highest accuracy, as well as the lowest MAE and MSE values.

Method	Acc (%)	MAE	MSE
BLNet	80.87	8.53	11.11
CCTrans	82.21	8.33	11.74
Ours-w48	82.33	7.49	8.7

## Data Availability

The Penaeus_1k datasets presented in this study can be found at the following link: [https://github.com/1L2018/plcs], (accessed on 1 June 2023).

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
