# Peer review of "Automatic Penaeus Monodon Larvae Counting via Equal Keypoint Regression with Smartphones"

_animals, 2023, doi:10.3390/ani13122036_

Round 1

Reviewer 1 Report

This paper discuss using smartphones to progress automatic counting on Penaeus monodon larvae. My comments are shown as below.

1. Please correct the reference format according to the journal formation.

2. Reference citations 4, 6, 7, 8, 9, 10 and 12 are all discussed the counting approaches for shrimp. Please mention the difference between your research and others.

3. In the paper title, the word 'smartphone' is involved. However, there is no related illustration about your smartphone. The camera efficient makes great influence about the captured images. Please add the description about the smartphone.

4. Some symbols illustrated in the article are incorrect, such as lines 266, 267and 274, equation (4). Please check the whole symbols used in the paper.

5. Definitions in Algorithm 1 - 3 and 4 are contradictory instructions. Candidate_points less than Th are expressed as 0, but which larger than 0 are expressed as 1. What's difference between candidate_points[ ] and candidate_points in [ ]? Do you mean the binarization operation? 

Algorithm 1 - 6, what is the meaning of '//'?

6. In Figure 6 (a), it is suggested the range of Acc can be adjusted from 70 to 100. In Figure 6 (b), please provide the equation of trend or erase it for no more related discussion in the paper.

7. In Figure 8,

(1) Please add the captions of two figures.

(2) In left figure, two other approaches show critical low accuracy on 53 and 183 groups. Please check the correctness or discuss them. In common sense, more numbers of Penaeus monodon should be presented the lower accuracy.

8. Please define the exact locations of three keypoints by image (line 424) or more exactly illustration.

9. What kind of HRNet is used in this study? There are several kinds of HRNets discussed in Reference 26.

The quality of English language is fine. However, some symbols mentioned in the article and before-and-after the equation are incorrected.

Reviewer 2 Report

attached as review report

it needs through revision for phrases mistakes and gramatical mistakes

Reviewer 3 Report

Dear authors,

I would like to congratulate you on the excellent work done in your research on counting tiger shrimp larvae using machine learning. The obtained results and the proposed approach have the potential to significantly contribute to the advancement of shrimp welfare and increased efficiency in shrimp farming. The practical application of your model can improve the accuracy of larval transactions between laboratories and shrimp farmers, as well as facilitate the planning of the breeding process.

However, I would like to alert you that, despite the remarkable work, I believe there is still room for necessary improvements. Therefore, some recommendations will be made to contribute to a more robust, relevant, and academically consistent final version of the work. These suggestions aim to improve the presentation of the results, deepen the analysis of variations in the dataset, and enhance the description of the model architecture.

By incorporating these recommendations, I believe your work will achieve even greater impact and relevance in the field of shrimp farming and artificial intelligence research. Once again, congratulations on the work already done and I wish you success in implementing the suggested improvements.

The text mentions two crowd counting methods that were compared to the proposed method for shrimp larvae counting: CSRNet and BLNet. The description of these methods is not in-depth in the text, as the main focus is the method proposed by the authors. However, as they serve as a basis for comparison to demonstrate the efficiency and accuracy of the proposed method, I suggest discussing these methods a bit more and even include more comparisons with other crowd counting methods and, if possible, specific methods for shrimp larvae counting, in order to strengthen the validation of the proposed method.

Discuss limitations: I suggest addressing the possible limitations of the proposed method better and providing a discussion on how they can be overcome in future work. This may include, for example, issues related to the generalization of the method to other species or challenges associated with variation in image quality, such as different lighting conditions and resolution, or the scalability of the method to high-density larval scenarios.

Describe the annotation process: Include more details on how the annotations were performed on the dataset, which tools were used, and whether any cross-validation was performed to ensure the quality of the annotations.

The authors do not provide specific details about the architecture of the proposed model. Without information about the model architecture, it is difficult to fully assess the effectiveness of the proposed method and how it outperforms other approaches.

The lack of information about the model architecture can be considered a deficiency in the article, as readers interested in replicating or adapting the proposed method in their own work may face difficulties in understanding the complete model design and how it works. Additionally, the lack of information about the model architecture makes it difficult to assess the originality of the proposed method compared to other existing approaches. It would be helpful if the authors provided additional information about the model architecture, including details about the layer structure, the activation functions used, optimization and loss function, as well as information about the model training and validation process. These details would help provide a more complete understanding of the proposed method and allow other researchers to evaluate and reproduce the results presented in the article.

The variation in the conditions of the dataset is an important aspect to consider when evaluating the performance of a machine learning model. These variations can include factors such as object density (in this case, shrimp larvae), lighting, resolution, camera angles, and occlusion. A robust model should be able to handle these variations and maintain good performance. In the article, the authors mention that the dataset has a rich diversity in terms of brightness, perspective, and resolution. However, it is important for the authors to provide additional information on how the model deals with these variations:

1)      Performance in different larval densities: The authors should analyze and discuss how the model behaves in different larval densities, including cases of low, medium, and high density. This will help to understand the robustness of the model in different conditions and its applicability in real-life scenarios.

2)      Lighting and image quality: The lighting conditions and image quality can vary significantly in natural environments. The authors should evaluate how the model deals with different lighting conditions, such as low contrast, shadows, glare, and excessive brightness. This would provide information on the model's ability to adapt to different environments and scenarios.

3)      Occlusion and camera angles: The presence of partial occlusions and different camera angles can affect the visibility of shrimp larvae in images. The authors should investigate the model's performance in situations of occlusion and different camera angles to determine its robustness in these conditions.

The authors do not directly address the scalability of the proposed model. Scalability is an important feature of a model, as it determines how well the method can be applied to larger or more complex datasets and how it can be adapted to solve similar problems in other domains.

To improve the description of the model's scalability, the authors could include additional discussions and analyses about the following aspects:

1)      Computational performance: The authors should describe how the model behaves in terms of training time, inference time, and required computational resources (e.g., memory and processing power). This would help readers understand if the model is efficient and if it can be easily applied to larger or more complex datasets.

2)      Generalization: The authors should provide information on how the model behaves in different scenarios and conditions, including variations in shrimp larvae density, lighting, resolution, and other image characteristics. This would help determine if the model can be successfully applied to other datasets and similar problems.

3)      Adaptation and transfer learning: The authors could discuss the model's ability to be adapted for other domains or related problems, leveraging the knowledge gained during training on the shrimp larvae dataset. This would include a discussion of how the model can be fine-tuned and which parts of the architecture can be reused.

Analyze the impact of variations in the dataset: Investigate how variations in the dataset conditions, such as lighting, position of the larvae, or image quality, affect the accuracy of the proposed method and discuss ways to deal with these variations.

The Quality of the English Language in the article is commendable, with the authors demonstrating a high level of proficiency in their writing. The text is well-composed, clear, and coherent, allowing readers to follow the arguments and understand the research findings presented quickly. The use of technical terms and jargon is appropriate and consistent, further contributing to the overall clarity of the text. Only minor grammatical or stylistic issues could be addressed to refine the language further, but these do not detract from the general high quality of the article's written expression.

Round 2

Reviewer 1 Report

1. Reference citation still needed to be revised, such as Ref. 19, 20, 21..., and so on.

2. The specification of capture device should be attached, such as solution or  pixels/inch.

Reviewer 2 Report

The article can be accepted for publication 

Author Response

We would like to thank you for your patience and time, as your detailed and comprehensive feedback during the review process has been instrumental to our progress. Your insights have not only helped us improve the content and structure of the manuscript, but have also contributed to our overall academic growth.